# Pilot Findings on SARS-CoV-2 Vaccine-Induced Pituitary Diseases: A Mini Review from Diagnosis to Pathophysiology

**DOI:** 10.3390/vaccines10122004

**Published:** 2022-11-24

**Authors:** Ach Taieb, El Euch Mounira

**Affiliations:** 1Department of Endocrinology, University Hospital of Farhat Hached Sousse, Sousse 4000, Tunisia; 2Faculty of Medicine of Sousse, University of Sousse, Sousse 4000, Tunisia; 3Laboratory of Exercice Physiology and Pathophysiology, Faculty of Medicine of Sousse, University of Sousse, Sousse 4000, Tunisia; 4Department of Internal Medicine, University Hospital of Charles Nicoles, Tunis 4074, Tunisia

**Keywords:** pituitary, SARS-CoV-2, COVID-19, vaccine, hypophysitis, apoplexy, ASIA syndrome, vaccine-induced thrombotic thrombocytopenia, VITT

## Abstract

Since the emergence of the COVID-19 pandemic at the end of 2019, a massive vaccination campaign has been undertaken rapidly and worldwide. Like other vaccines, the COVID-19 vaccine is not devoid of side effects. Typically, the adverse side effects of vaccination include transient headache, fever, and myalgia. Endocrine organs are also affected by adverse effects. The major SARS-CoV-2 vaccine-associated endocrinopathies reported since the beginning of the vaccination campaign are thyroid and pancreas disorders. SARS-CoV-2 vaccine-induced pituitary diseases have become more frequently described in the literature. We searched PubMed/MEDLINE for commentaries, case reports, and case series articles reporting pituitary disorders following SARS-CoV-2 vaccination. The search was reiterated until September 2022, in which eight case reports were found. In all the cases, there were no personal or familial history of pituitary disease described. All the patients described had no previous SARS-CoV-2 infection prior to the vaccination episode. Regarding the type of vaccines administered, 50% of the patients received (BNT162b2; Pfizer–BioNTech) and 50% received (ChAdOx1 nCov-19; AstraZeneca). In five cases, the pituitary disorder developed after the first dose of the corresponding vaccine. Regarding the types of pituitary disorder, five were hypophysitis (variable clinical aspects ranging from pituitary lesion to pituitary stalk thickness) and three were pituitary apoplexy. The time period between vaccination and pituitary disorder ranged from one to seven days. Depending on each case’s follow-up time, a complete remission was obtained in all the apoplexy cases but in only three patients with hypophysitis (persistence of the central diabetes insipidus). Both quantity and quality of the published data about pituitary inconveniences after COVID-19 vaccination are limited. Pituitary disorders, unlike thyroid disorders, occur very quickly after COVID-19 vaccination (less than seven days for pituitary disorders versus two months for thyroid disease). This is partially explained by the ease of reaching the pituitary, which is a small gland. Therefore, this gland is rapidly overspread, which explains the speed of onset of pituitary symptoms (especially ADH deficiency which is a rapid onset deficit with evocative symptoms). Accordingly, these pilot findings offer clinicians a future direction to be vigilant for possible pituitary adverse effects of vaccination. This will allow them to accurately orient patients for medical assistance when they present with remarkable symptoms, such as asthenia, polyuro-polydipsia, or severe headache, following a COVID-19 vaccination.

## 1. Introduction

Coronavirus disease 2019 (COVID-19) has been initially labeled as a severe potentially lethal respiratory infection caused by the severe acute respiratory syndrome coronavirus 2 (SARS-CoV-2) [1]. However, it has slowly become clear that many extra-pulmonary manifestations greatly contribute to the severity of the infection [2,3]. Furthermore, COVID-19 seems to have a wide-ranging impact on the body with little-known clinical features [4]. Data relating to the virus’ impact on the endocrine system is gaining momentum by the day. It has become an increasingly studied subject that sheds light on a variety of disorders, such as pituitary lesions, thyroid dysfunction, diabetes, adrenal insufficiency, and hypogonadism [5,6,7]. There is evidence that organs expressing the angiotensin-converting enzyme 2 (ACE2) receptor (such as the pituitary gland, thyroid, pancreas, adrenals, and gonads) can be the targets of the virus, as this receptor enhances the SARS-CO-2 attachment, permitting the infliction of cell damage [8,9]. COVID-19 infection can disrupt the functioning of endocrine organs and, similarly, COVID-19 vaccines can also induce endocrine dysfunction [5,10,11]. Multiple vaccines with variable efficacy and safety have been developed against COVID-19. Both the effect of the vaccine and the clinical course of the infection are inherently related to the systemic physiological response of the body. They remain also conditioned by the underlying health status of the patients [12]. The large use of vaccines worldwide (Pfizer–BioNTech, Oxford–AstraZeneca, Sinopharm BIBP, Moderna, Janssen, CoronaVac, Covaxin, Novavax, Convidecia, and Sanofi–GSK), allows for a broad evaluation of their side effects [13]. To date, the documented endocrine dysfunctions following COVID-19 vaccination concern primarily the thyroid gland, the pancreas’ beta cells, the adrenal glands, and the pituitary glands [5,7] In the reviewed data, the most commonly reported thyroid dysfunction was subacute thyroiditis, followed by Graves’ diseases [14,15]. The onset of symptoms, such as neck swelling or clinical thyrotoxicosis, ranged from 4 to 21 days post-vaccination. Certain series in the literature also noted cases of type 1 diabetes uncovered three to five weeks after vaccination and for which the patients received antidiabetic medication and remained well controlled for several weeks after discharge [16,17]. Additionally, disorders related to the adrenal gland have been reported, namely bilateral adrenal hemorrhage and primary adrenal insufficiency [18]. Pituitary dysfunctions seem to be among the rarest side effects recorded in the literature. Only a few case reports have been identified revealing a wide variety of clinical presentations [5]. Unlike signs of dysthyroidism or the onset of diabetic ketosis, SARS-CoV-2 vaccine-induced pituitary diseases can take on clinical presentations that are intertwined with a classical post-vaccination syndrome (headache, asthenia...) [5]. This further explains why some cases of pituitary-induced diseases might have gone undetected, especially if the clinician did not suspect and investigate their presence. Over the last few months, an increasing number of papers discussing the repercussions of SARS-CoV-2 vaccines and their management have urged the need for a thorough pilot analysis of all the materials published so far, which we believe could be of great value for endocrinologists and clinical practitioners.

## 2. Materials and Methods

The search strategy through PubMed/MEDLINE included original papers, case reports, and case series articles reporting pituitary disease following SARS-CoV-2 vaccination that were published until September 2022. A total of 8 cases were collected, among which are letter to the editors and case reports. The search terms, used both separately and in combination, included the following: “SARS-CoV-2”, “COVID-19”, “hypophysitis”, “pituitary stalk thickness”, “hypopituitarism”, “autoimmune pituitary disease”, “vaccine”, “vaccination”, “pituitary insufficiency”, and “diabetes insipidus”.

Pituitary insufficiencies were diagnosed using basal hormone concentrations and/or dynamic hormone tests. Panhypopituitarism was defined as three or more of pituitary hormone deficiencies [19,20]. Magnetic resonance imaging was performed in all subjects.

Serum free thyroxine, thyroid-stimulating hormone TSH, prolactin, luteinizing hormone (LH), follicle-stimulating hormone (FSH), cortisol, growth hormone (GH), insulin-like growth factor 1 (IGF-1), estradiol, and testosterone concentrations were analyzed in all of the cases. Dynamic hormonal tests, including an adrenocorticotrophic hormone (ACTH) stimulation test and a water deprivation test, were performed in some cases.

Norms for pituitary insufficiencies were defined as follows: For TSH deficiency, diagnosis was made based on low basal serum thyroxine with an inappropriately normal or low TSH [21]. Low serum testosterone in men and low serum estradiol in pre-menopausal women, along with an inadequately low or normal gonadotrophin levels, defined Gonadotrophin deficiency [20]. Baseline and dynamic testing were used to diagnose ACTH insufficiency. In untreated patients, a basal (9 am) cortisol lower than 138 nmol/l confirmed the deficiency. However, dynamic testing was performed in patients with serum levels between 100 and 450–500 nmol/L. Peaks greater than 450–500 nmol/l obviated the need for provocative tests of ACTH reserve [22]. Assessment of GH deficiency was made through the evaluation of IGF-1 concentration, which was abnormally low [23]. Provocative tests were not used in these cases. Antidiuretic hormone (ADH) deficiency confirmation was made using an 8-h water deprivation test [24]. Patients were denied fluids for 8 h, or until 5% of their body mass had been lost. During and after the test, plasma and urine osmolalities, along with urine volumes and body weight, were estimated every two hours. At the end of the 8 h deprivation, 2 μg of desmopressin were injected. In patients with central diabetes insipidus (DI), the urine, diluted at the end of fluid deprivation (less than 300 mOsmol/kg), became concentrated (greater than 750 mOsmol/kg) following the desmopressin injection [25]. An additional radiological feature for the diagnosis of DI was retained, consisting in the loss of posterior pituitary bright spot on T1 weighted Magnetic Resonance Imaging (MRI) [26]. While all axes were seen downward, only prolactin might rise in the case of stem damage or inflammation. Hyperprolactinemia was defined as a prolactinemia that was above the upper normal limit of serum level in most laboratories (15 to 20 ng/mL) [27].

## 3. Results

Eight cases were included [28,29,30,31,32,33,34,35]. Regarding the type of vaccines administered, 50% of the patients received (BNT162b2; Pfizer–BioNTech, New York, NY, USA) and the other 50% received (ChAdOx1nCov-19; AstraZeneca London, UK). In five cases, the pituitary disorder appeared after the first dose of the corresponding vaccine [28,29,30,33,34]. In all cases, there were no personal or familial history of pituitary disease described. All the patients described had no previous SARS-CoV-2 infection prior to the vaccination episode. No other side effects related to previous vaccination had been notified. All these cases have been summarized in Table 1. We found five cases of vaccination-induced hypophysitis and three cases of pituitary apoplexy.

### 3.1. Hypophysitis

#### 3.1.1. Clinical Presentations

Post-vaccination inflammatory pituitary disease was the most common condition in five patients. The ratio was in favor of women (3/2). The onset was quite fast, ranging from one day to a maximum of seven days post-vaccination. Among these patients, four patients had received the Pfizer–BioNTech BNT162b2 vaccine [31,32,33,35] versus a single patient with the AstraZeneca ChAdOx1 nCov-19 vaccine [34]. All hypophysitis cases presented symptoms suggestive of a hypothalamic–pituitary involvement. The classic presentation of hypophysitis does not differ from that encountered in patients with post-vaccinal involvement. Among them, we found headaches, which were the most frequent symptom in almost all of the patients. The tumoral syndrome caused by hypophysitis might be the cause of these and other symptoms, in particular visual blurring. Hypophysitis, apart from sellar, infra- and supra-sellar tumor syndromes, also causes hormonal deficits that we found in these patients. Anterior pituitary involvement was present in three cases [31,33,35]. Two patients had hypopituitarism in three axes (corticotroph, gonadotroph, and thyreotroph), indicating the intensity of the inflammation [33,35]. One patient had only isolated corticotropic involvement, which was evoked by the symptoms of acute corticotropic insufficiency (nausea, vomiting, and abdominal pain) [31]. The most interesting in these cases was post-pituitary damage which was the most frequent, with four out of five patients being affected [32,33,34,35]. Post-pituitary involvement was evoked by the typical polyuro polydipsic syndrome found in all patients. The craving for water reached up to 8 L in some patients, and the sudden onset raised suspicion of acute ADH deficiency (after eliminating hyperglycemia in particular). All patients had a water restriction test which showed an initial absence of urine concentration, hypotonic urine, and positive free water clearance. These parameters normalized with the addition of desmopressin which made it possible to concentrate urine, confirming the diagnosis of ADH deficiency. All patients received desmopressin at discharge, together with the treatment of other pituitary deficits.

#### 3.1.2. MRI Findings

When pituitary deficit was confirmed, the first-line examination that was performed on all of the studied patients was a hypothalamic–pituitary MRI. A thickening of the pituitary stalk was found in two patients [33,34], who had particularly symptomatic central DI with an intense polyuro-polydipsic syndrome. Loss of signal and enlargement of the pituitary were found in only one patient [32]. Pituitary atrophy found in a single patient was revealed by isolated corticotropic insufficiency [31].

#### 3.1.3. Treatment

All patients received hormonal replacement for the deficits of the affected axes. For ADH deficiency, treatment was based on oral Desmopressin. Treatment of hypophysitis involved high-dose corticosteroid therapy. The follow-up was favorable in the case of Murvelashvili et al. [35], and in other reported cases where the imaging aspect of hypophysitis regressed at one month [31,32,33]. In the case of Ach et al., the appearance on the imaging, as well as the polyuro-polydipsic syndrome, persisted beyond three months [34].

### 3.2. Pituitary Apoplexy

#### 3.2.1. Clinical Presentations

Pituitary apoplexy (three cases) was only associated with the ChAdOx1 nCov-19 vaccine; AstraZeneca [28,29,30]. The disorder was observed only in female patients. The clinical presentation was revealed after one day of vaccination in two patients [28,29], and in five days in another one [30]. Symptoms followed the first injection of the vaccine in all the cases.

The clinical presentation was highly symptomatic in all patients, mainly with headaches varying in intensity and fever in one patient. Headaches were characteristic of pituitary involvement since they were retro-orbital and frontal. The symptoms were associated with hemodynamic disorders in some patients and asthenia. Pituitary lesions were uncommon. Only one patient had a hormonal deficit in the gonadotropic axis [29]. There was no corticotropic deficiency or ADH deficiency. Hyperprolactinemia was found in only one patient [29]. The resolution of the clinical picture varied from patient to patient, possibly taking several weeks depending on the initial severity. Analyses of D-Dimer and complete blood count were not performed in all patients. Some of them showed normal D-dimer levels [29].

#### 3.2.2. MRI Findings

In all patients, the pituitary gland was enlarged. A mass effect was found, with sellar, chiasmatic, and cavernous sinus compressions. There was an alteration of the pituitary signal in one patient [29]. In another patient, a suspected pituitary adenoma was found [30].

#### 3.2.3. Treatment

The patients received hormonal replacement for pituitary deficiencies, similar to the patients with hypophysitis. Analgesics were also used. Clinical improvement took time in three cases, with some requiring a follow-up of more than three months to recover a healthy pituitary aspect on imaging. In one case, the involvement of the posterior pituitary persisted for several months [29].

## 4. Discussion

An unprecedented acceleration of vaccine development was observed amidst the COVID-19 pandemic, mainly gene-based and inactivated vaccines with variable efficacy and safety profiles [13]. The mRNA-based vaccines are comprised of an RNA protected by lipid nanoparticles. Once delivered into the host cells, the RNA encodes the SARS-CoV-2 spike protein. This transient expression of the viral spike glycoprotein S is responsible for eliciting a protective immune response [36,37,38]. However, the inactivated adenovirus-based SARS-CoV-2 vaccines are designed to act as vectors for the full-length viral genes. They infect the cells to deliver the genes, but do not replicate [39]. The inactivated vaccines are bioprocessed through vero cell lines [40]. They are chemically inactivated by β-propiolactone and then adjuved with aluminum hydroxide to boost the response of the immune system [41]. This adjuvant, however, has led to some rare but serious adverse events, including anaphylaxis, thrombotic events and thrombocytopenia, Guillain–Barre syndrome, myocarditis, and even death [42].

Amidst the ongoing SARS-CoV-2 vaccination campaign, a large spectrum of endocrine dysfunctions was reported in the literature involving the thyroid gland, pancreatic islets, the pituitary, and the adrenal gland [5]. So far, many reviews have focused on the analysis of thyroid or pancreatic disorders [14,17]. The cases of pituitary disorders are not very numerous, but we strongly believe that they are underestimated, given the intricacy of the clinical symptoms with features of COVID-19 itself. Pituitary disorders were more common among women in this analysis. The two most preponderant disorders that we collected were hypophysitis and pituitary apoplexy. Their clinical presentation was slightly different, but they mainly diverged on pathophysiological aspects, which we will clarify further in this mini review and pilot findings.

### 4.1. Hypophysitis

#### 4.1.1. Pathogenesis

Hypophysitis is a heterogeneous condition attributable to an inflammation of the pituitary gland and its stalk. Inflammation can affect only the anterior pituitary, the posterior pituitary, or the entire gland resulting in panhypophysitis [43]. Hypophysitis can be broadly classified into primary and secondary forms. The precise etiology of primary hypophysitis remains unknown. It amounts to autoimmune and other inflammatory or infiltrative forms of isolated pituitary involvement [44]. Secondary hypophysitis occurs as a result of a reaction of the pituitary gland to local or systemic events, among which are local processes, systemic diseases, infections, neoplastic processes, and drugs [45]. Apart from etiological classification, hypophysitis can be presented histologically with various aspects. It has lymphocytic, granulomatous, xanthomatous, IgG-4 related, necrotizing, and mixed forms [43]. However, accurate classification is not always possible due to the overlap in pathological features.

The presentation of hypophysitis induced by systemic inflammatory diseases does not differ from those induced by anti-COVID-19 vaccines. However, the pathophysiological mechanisms evoked are different. Several of these mechanisms cannot be proven directly since it is necessary to carry out a pathological analysis of the pituitary gland to allow this confirmation [46]. Since the pituitary is a gland that is very difficult to access for biopsy, performing such acts can lead to complications, particularly an aggravation of pre-existing deficits. This failure rate of biopsy is even higher in the event of isolated damage to the pituitary stalk [47].

Among the pathophysiological mechanisms of post-vaccine hypophysitis, autoimmune and inflammatory syndromes induced by vaccine adjuvants (ASIA) were the most cited [48].

Adjuvants are substances used to enhance the magnitude and durability of the immune response, which makes them an undeniable asset in the upgrading processes of vaccines. In genetically susceptible subjects, exposure to adjuvants may, on rare occasions, set off polygenic auto-immune diseases [49]. The immune disruption in such cases is attributed mainly to molecular mimicry, which triggers polyclonal activation of B lymphocytes [50]. The reviewed data on adjuvants used in COVID-19 vaccines showed that components such as aluminum salts, emulsions, oils, toll-like receptors, AS01B, four lipids of the mRNA vaccine, and polyethylene glycol might generate an immune response in susceptible individuals (Figure 1) [38,40,49].

Classically, many endocrine dysfunctions (e.g., type 1 diabetes mellitus, primary ovarian failure, adrenal insufficiency, and thyroiditis) have been associated in the literature with the ASIA syndrome following viral infections, such as human papillomavirus and hepatitis B virus, and even after influenza vaccination [51]. Similarly, these endocrinopathies were reported in association with the ASIA syndrome in a published review by Bargazzi et al., along with hypophysitis [52].

From May 2021 to December 2021, the 36 patients referred to hospitals in Mexico City were diagnosed with the ASIA syndrome after immunization with different types of COVID-19 vaccines [48]. The data regarding this entity remain, however, limited and are subjected to debate. Three major criteria define the ASIA syndrome: First, there is an exposure to an external stimulus (such as COVID-19 vaccine). Second, there is an acute onset of clinical features of autoimmune diseases (including vasculitis, arthritis, and neurological syndromes), which has been noted shortly after a COVID-19 vaccine injection. Third, improvement occurs after the removal of the inciting agent (this criteria is evidently irrelevant for vaccines) [53].

Another pathophysiological mechanism arises that implies a hyper-stimulation of the immune system and molecular mimicry of the vaccine components [54]. Interestingly, SARS-CoV-2 proteins (from its spike, nucleus, and membrane) present sequence with similitude to some peptides in endocrine tissues, such as peroxidase in the thyroid gland [55]. Autoimmune inflammation results, therefore, from a cross-recognition between the modified SARS-CoV-2 proteins in vaccines and the human target protein [36]. In the majority of cases, symptoms of hypophysitis—attesting of auto-immune activation—appear a few days after vaccination. The concentration of viral proteins, peaking synchronously, could be the underlying reason. In fact, the vaccine-encoded antigen (S protein) is stabilized in its perfusion form in the mRNA vaccines. It spreads systemically through circulation and, therefore, could contribute to the triggering of autoimmune diseases in susceptible individuals [55].

Following adjuvant internalization and mRNA release, the viral signal peptide drives antigen production in the endoplasmic reticulum (ER). After sorting in the Golgi network, S protein acquires its final position in the human cell membrane, where S1 is exposed to the extracellular space. Antigen sorting and trafficking may also induce the release of S protein-containing exosomes. Figure 2 also shows dendritic cells (professional antigen-presenting cells) engulfing circulating antigens and antibody-mediated binding of B cells to cell-anchored antigens. All the above-mentioned mechanisms potentiate the inflammatory mechanisms. The endocrine consequences after hypophysitis are mainly hypogonadism, hypothyroidism, hypoadrenalism, and diabetes insipidus.

As for the last mechanism, we discuss the implications of a systemic inflammatory response. Cell damage inflicted by SARS-CoV-2 can result in many aberrations. It can cause cell degeneration [10]. It might prompt a pro-inflammatory cytokine reaction that directly results in compromised insulin receptor signaling and in islet cell damage [56]. Furthermore, it could potentially increase renin–angiotensin system activation via an ACE receptor downregulation, subsequently impairing the pituitary’s receptor signaling [8].

Thus, it is reasonable to suppose that SARS-CoV-2 antigen presentation, occurring through vaccination, can also trigger similar responses. We are inclined to think that the DI in these patients was due to SARS-CoV-2 vaccine-induced hypophysitis because we observed a clear sequential relationship between SARS-CoV-2 vaccine administration and the onset of hypophysitis, along with an exclusion of the common causes of hypophysitis [57].

These impairments were noted in patients without a notable history of previous COVID-19 infection that could have explained these lesions. These findings were also reinforced by the concordance between the clinical and chronological aspects and the pharmacovigilance and drug safety information.

#### 4.1.2. Clinical Presentations

Hypophysitis typically presents with symptoms that can be attributed either to pituitary deficiencies or to the mass effect of an enlarged pituitary gland and infundibulum. Symptoms related to the mass effect are quite common: visual disturbances concern 10% to 30% of patients with primary hypophysitis, while headaches affect approximately half of them [43]. The etiology, severity, and extent of the pituitary damage are the main factors dictating the loss of anterior and/or posterior pituitary hormones and, therefore, conditioning the clinical presentation. For instance, when anterior pituitary dysfunction is associated with DI, the damage is most likely due to an inflammatory process or to metastasis, as this clinical presentation is highly improbable in pituitary adenomas [44]. In vaccine-induced hypophysitis, inflammatory processes predominantly and equally affect corticotrophs, gonadotrophs, and thyreotrophs. Most importantly, this pattern is similar to other classical hypophysitis, unlike in pituitary adenomas which rarely affect the corticotroph axis [58]. All patients require a complete pituitary function testing, including the following: an 8 am cortisol and/or adrenocorticotropin stimulation test; TSH; free thyroxine; prolactin; IGF-1; LH; and FSH with estradiol in premenopausal women or testosterone in men. If DI is suspected, the levels of serum sodium should be determined along with plasma and urine osmolarity. Hyponatremia caused by adrenal insufficiency or severe hypothyroidism should be ruled out. Depending on the extension of the inflammation and the clinical context, each case may have systemic features related to the severity of tissue damage [45]. In the studied cases, clinical presentations varied from a single axis deficiency in one patient (isolated corticotroph deficiency) to multiple pituitary deficits.

Numerous neuro-radiological findings have been reported in vaccine-induced hypophysitis [59]. The main features suggestive of the disease include a diffuse and symmetric mild-to-moderate pituitary gland enlargement and/or a thickened, non-deviated stalk. Contrast uptake is usually intense and generally homogeneous [60]. In infundibulo-neurohypophysitis, stalk thickening is isolated (without features of gland enlargement); the loss of posterior pituitary bright spot can also be seen and is attributed to the depletion of vasopressin granules [24]. Images of an enlarged pituitary gland may mimic a pituitary adenoma, leading to misdiagnosis [60]. Contrariwise, in advanced stages of the disease, an atrophic aspect of the pituitary gland can be observed along with images of sellar arachnoidocele or empty sella [61].

Despite lacking the specificity for diagnosing vaccine-induced hypophysitis, MRI findings can help establish a diagnosis in the right clinical context with a major benefit of being a noninvasive method. For example, proof of stalk thickening in a woman who has recently been vaccinated and is presenting with headache and hypopituitarism is highly suggestive of hypophysitis [34].

#### 4.1.3. Treatment

Owing to their anti-inflammatory properties, glucocorticoids (GCs) are regarded as the treatment cornerstone of hypophysitis [44]. In the chronic phases, however, when irreversible damages take place, anti-inflammatory treatment may not affect radiological or hormonal outcomes. Moreover, spontaneous resolution of pituitary infiltration with or without permanent pituitary dysfunction did occur in some cases [43]. In vaccine-induced hypophysitis, the evolution remains unclear. Occasionally, the process persisted up to several months, requiring a regular follow-up [34]. In a German cohort, radiological improvement was noted in 46% of patients with primary hypophysitis who were managed through observation only. Additionally, one third of the patients had hormonal recovery, mainly including vasopressin and ACTH [62].

It is still uncertain whether GCs are superior to simple observation in allowing for a better pituitary function recovery or not, as no randomized controlled studies have been performed yet. Given the broad spectrum of GC side effects, a consideration of the risks and benefits is crucial when deciding whether to actively treat mild cases of primary hypophysitis or not. Clinical signs and symptoms should be taken into consideration when selecting a treatment option. For example, in patients with mild-to-moderate headache, mild pituitary dysfunction, and no mass effect on optic chiasm, observation may be safely envisaged [43,44]. Hormonal replacement of the other pituitary axes is also necessary.

Hormonal replacement doses in vaccine-induced hypophysitis do not differ from other pituitary diseases in the reported cases. Essentially, for DI, vasopressin replacement should be maintained in case of persistent polyuro polydipsia. Patient education remains important in order to avoid a decompensation of these hormonal deficiencies.

Tumoral compression on neighboring structures is rarer in hypophysitis than in the case of pituitary apoplexy [63]. In such cases, glucocorticoids will seemingly decrease the inflammatory process. In addition, a surgical intervention could be considered in order to outperform a debulking to save the optic chiasma. The followed patients did not mention any mechanical compression that could compromise the visual prognosis or the integrity of cranial nerves.

### 4.2. Pituitary Apoplexy

#### 4.2.1. Pathogenesis

The Oxford–AstraZeneca COVID-19 vaccine is a viral vector vaccine using the modified adenovirus ChAdOx1. The vaccine contains a full-length coding sequence of SARS-CoV-2 spike protein in the form of DNA. When the viral vectors infect the host cells, the aforementioned genetic code is transcribed, translated, and then presented as a spike protein to the body’s immune system by the host cells. The ChAdOx1 nCov-19 vaccination produced a positive immune reaction in all of the patients with apoplexy [64]. Following several large vaccine campaigns, there was mounting evidence of serious adverse effects related to this vaccine, including thrombosis and bleeding. This syndrome has been termed “Vaccine-Induced Thrombotic Thrombocytopenia” (VITT) [65]. It occurred more frequently in young women and was imputed only to viral vector vaccines. VITT seems to be a phenomenon similar to heparin-induced thrombocytopenia and appears to have an autoimmune provenance [66]. Currently, five criteria are used to define VITT, including recent vaccination, thrombosis, thrombocytopenia, elevated D-dimer levels, and positivity of anti-PF4 antibodies (Figure 3) [67].

However, the mandatory presence of all five criteria in order to confirm a diagnosis of VITT remains a subject of debate. It could exclude patients in the “grey zones” of probable or possible VITT. By way of illustration, adopting a strict cut-off for thrombocytopenia (150 × 10^9^/L) could exclude patients with sufficient evidence of VITT, who present with typical or even worrisome symptoms. A correct analysis of vaccine-induced apoplexy is difficult since there are numerous limiting aspects [68]. For example, the patients included in this paper had some degree of discrepancy when defining VITT, although thrombosis and thrombocytopenia were mentioned in one case. Additionally, the authors did not provide sufficient clinical evaluation of severity. The main hypotheses, explaining post vaccine apoplexy, are inflammatory or immune reactions leading to endothelial dysfunction. The latter results in hyper permeability, which increases the risk for hemorrhages and cerebrovascular events [69]. Both the extensiveness and the fragility of the vascular network of the pituitary gland make it very vulnerable to this type of reactions [70]. An increase in pituitary volume by subsequent edema or hemorrhage will increase the mechanical compression of the gland and contribute to reducing the blood supply to the pituitary [71]. All these elements will aggravate pituitary ischemia and worsen the prognosis of apoplexy by accelerating glandular necrosis. Thus, we suggest that patients could have an underlying pituitary hypophysitis that could have developed an immunological reaction after being triggered by the COVID-19 vaccination, resulting in an apoplexy.

#### 4.2.2. Clinical Presentations

The most prevalent symptom of pituitary apoplexy is a headache of severe and sudden onset, located behind the eyes. The proposed mechanisms to explain its origin include involvement of the superior division of the trigeminal nerve inside the cavernous sinus, meningeal irritation, dura-mater compression, and enlargement of sellar walls [72]. The headache can be associated with other clinical manifestations, such as decreased visual acuity, hemianopsia, ptosis, nausea and vomiting, altered mental status, and hormonal dysfunction. Patients can also experience diplopia, or double vision, when extrinsic compression of one or several extraocular nerves occurs [73]. In the context of pituitary apoplexy, the oculomotor is the most commonly affected nerve. Damage to this nerve could result in ptosis and lateral eye deviation, sometimes accompanied by a pupillary dilation of the affected eye [73]. More than two-thirds of the patients with apoplexy present with ACTH deficiency [74]. This is alarming as the subsequent cortisol depletion can be life threatening for patients. They may experience an adrenal crisis with various symptoms, such as nausea and vomiting, abdominal pain, bradycardia and hypotension, hypothermia, lethargy, and sometimes coma.

A head CT scan, performed as a first-line imaging exam, shows a sellar/suprasellar mass associated with intralesional hemorrhage [72,75]. It cannot, however, identify ischemia or necrosis of the gland/tumor. Contrast examination, which was done afterwards, helps delimit the size of the tumor through the evaluation of contrast-enhanced areas which appear hyperdense in the brain [76]. Brain MRI is the gold-standard imaging exam in diagnosing apoplexy. Through precise delimitation of hemorrhagic and necrotic areas, it improves the definition of the lesions [77]. Radiological features differ in ischemic and hemorrhagic apoplexy. In ischemic pituitary apoplexy, brain MRI images objectify an enlarged sellar/suprasellar mass, with peripheral enhancement surrounding a hypointense center [78]. Brain MRI also provides information about the consistency of the tumor and the exact topography of ischemic tissues via diffusion-weighted imaging [76]. On the other hand, in hemorrhagic apoplexy, the presence of blood translates to intralesional areas of high signal intensity within the sellar/suprasellar lesions seen on T1-weighted MRI [72]. Moreover, gradient-echo sequences in MRI, notably T2* weighted, are very sensitive in detecting deposits of hemosiderin [72].

#### 4.2.3. Treatment

Pituitary apoplexy, either hemorrhagic or ischemic, is considered a medical emergency. Treatment requires a multidisciplinary approach (including neurologists, neurosurgeons, and intensive care teams), and is mainly conditioned by the severity of the clinical presentation [63,73]. Patients may require immediate medical assistance, including a careful assessment of fluids and electrolyte balance, to ensure hemodynamic stability [71]. Hormonal disturbances, namely cortisol and thyroid hormone deficiencies, should be urgently corrected. GC administration is to be prioritized and should not be delayed by investigations. It concerns all patients even in the absence of signs of adrenal crisis. The recommended dose is an intravenous 100–200 mg bolus of hydrocortisone. This bolus should be followed either by additional administration of 50–100 mg every 6 h, or a continuous intravenous infusion of 2–4 mg/hour [76]. Thyroid hormone replacement, if required, should be prescribed only after glucocorticoid administration [71].

Both conservative and surgical treatments are considered for the management of pituitary apoplexy. In the reviewed literature, outcomes were assessed according to these two types of intervention (conservative vs. surgical management). Some studies further evaluated outcomes based on the timing of surgery (early vs. delayed) or its modalities (microscopic vs. endoscopic transsphenoidal resection) [79]. It was noted that surgical treatment resulted in a rapid alleviation of headache. However, the recovery time of other sequelae of pituitary apoplexy was variable [80]. Conservative treatment strategies were reserved for mild and stable conditions of apoplexy [71]. There are no clear data on how to manage patients once they are stabilized. Nonetheless, reiterated clinical and radiological surveillance is recommended [81].

For patients with VITT, prompt treatment with high-potency glucocorticoids and intravenous immunoglobulin was found to efficiently improve platelet counts, resulting in lower hemorrhagic risks [67,82]. Treatment options also include plasma exchange which can be performed in critical situations. This temporarily reduces the level of anti-PF4 antibodies and is reserved for resistant forms of the disease [83]. Platelet transfusions, however, should be proscribed in individuals with VITT since they considerably increase the risk of thrombotic events [82].

In this review, the selected patients showed great recovery after steroid treatment and, therefore, plasma exchange was not needed.

## 5. Conclusions

The safety data on COVID-19 vaccines seem reassuring, and there is no way to deny the value of this immunization in view of its importance in reducing severe forms of COVID-19 infections. The cases of endocrinopathies reported in relation to these vaccines are only scattered cases. Unlike thyroid or pancreatic impairment, pituitary damage is much rarer and plausibly under-diagnosed. This is mainly due to the entanglement of its symptoms with those of post-vaccination flu syndrome or those of a potential long COVID. Damage in the pituitary gland can be due either to hypophysitis or to pituitary apoplexy. The physiopathology, however, does not seem to differ enormously since it involves the ASIA syndrome which can be a consequence of the involvement of either the adjuvants or the Spike S proteins. The hypothesis of the VITT syndrome is also mentioned in the case of pituitary apoplexy. With that said, it is important for clinicians to know the main symptoms evoked by these disorders. These features include retro-orbital headaches with visual blurring, polyuro-polydipsic syndrome (in posterior pituitary dysfunction), and deterioration in the general state with great asthenia (in the case of corticotroph damage). Both corticotroph damage and diabetes insipidus can be life threatening and should alarm practitioners. The presence of these symptoms in the immediate post-vaccination period urges the need for clinicians to explore hormonal axis in search for pituitary deficits, which should be urgently and accordingly treated.

## Figures and Tables

**Figure 1 vaccines-10-02004-f001:**
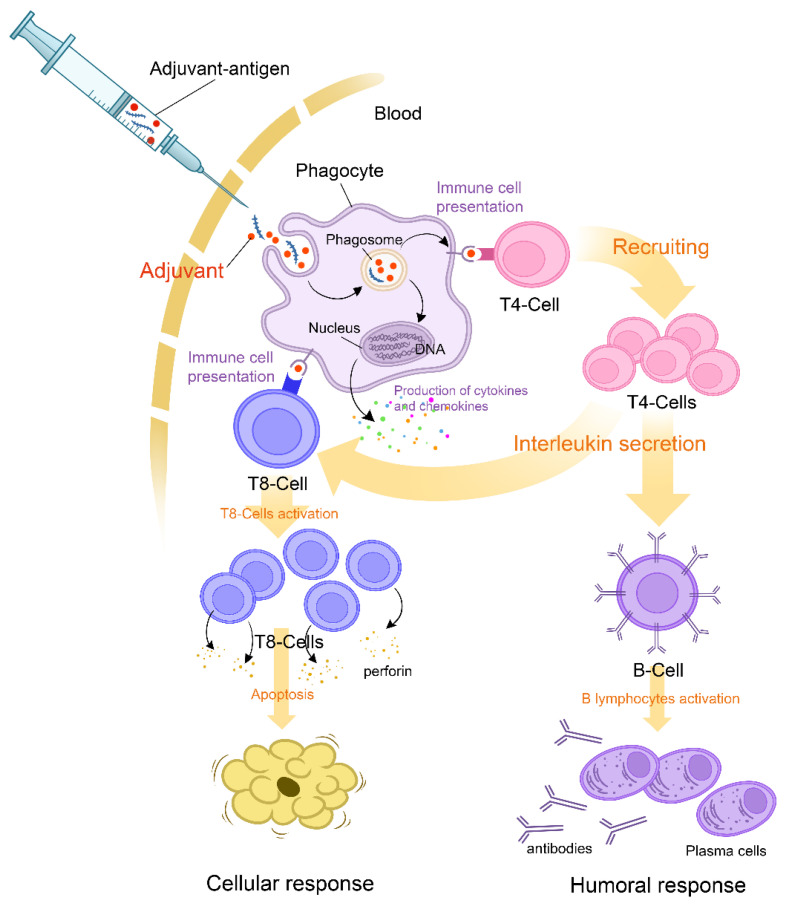
Mechanism of action of adjuvants and initial triggers explaining the pathophysiology of the ASIA syndrome following COVID-19 vaccination.

**Figure 2 vaccines-10-02004-f002:**
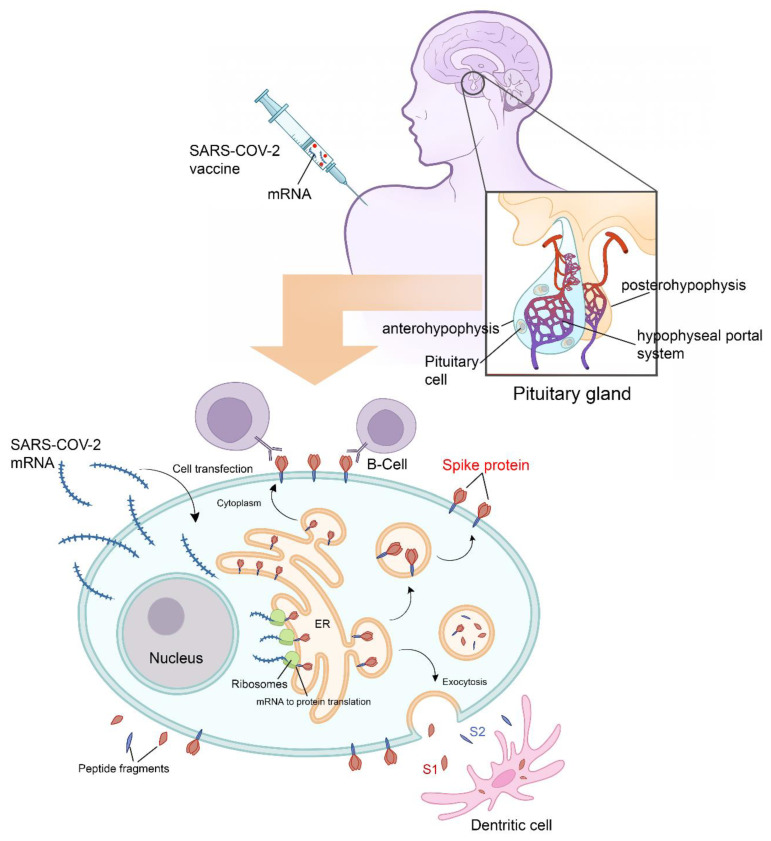
Physiology of pituitary cell protein’s expression and vaccine-induced hypophysitis pathophysiology. Following adjuvant internalization and mRNA release, the viral signal peptide drives antigen production in the endoplasmic reticulum (ER). After sorting in the Golgi network, S protein acquires its final position in the human cell membrane, where S1 is exposed to the extracellular space. Antigen sorting and trafficking may also induce the release of S protein-containing exosomes. Also shown are dendritic cells (professional antigen-presenting cells) engulfing circulating antigens, and antibody-mediated binding of B cells to cell-anchored antigens. All the above mentioned mechanisms potentiate the inflammatory mechanisms. All the Endocrine consequences after hypophysitis are mainly hypogonadism, hypothyroidism, hypoadrenalism and diabetes insipidus.

**Figure 3 vaccines-10-02004-f003:**
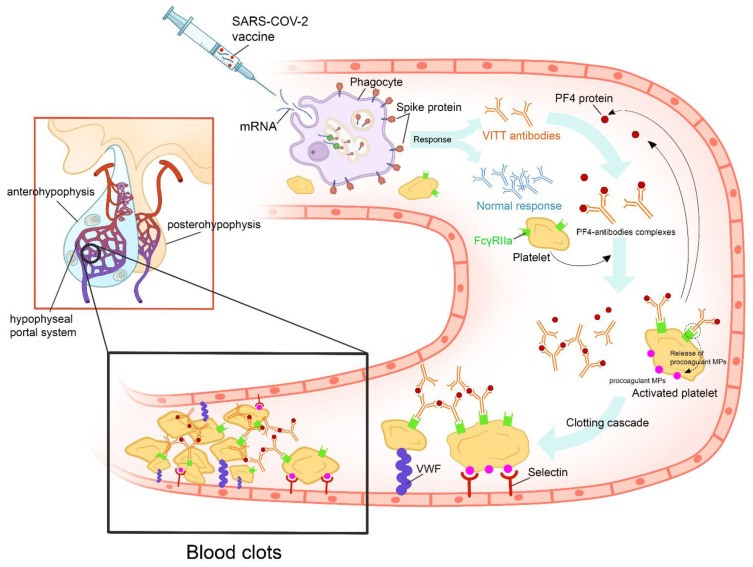
Proposed figure showing the cascade of pathogenic events that could favor the onset of vaccine-induced immune thrombotic thrombocytopenia in individuals vaccinated with anti-COVID-19 vaccines. After the administration of the vaccine, the recipient’s cells produce harmless COVID-19 proteins and the immune system responds by producing protective antibodies. In some cases, the vaccine’s adjuvants and spike proteins trigger a type I interferon response and the production of VITT antibodies. VITT is caused by antibodies that recognize PF4 bound to platelets. These antibodies are IgGs that activate platelets via low-affinity platelet FcγRIIa receptors (receptors on the platelet surface that bind the Fc portion of IgG). Anti-PF4 antibodies cause cellular activation that, besides activating platelets and coagulation reactions, activates monocytes, neutrophils, and endothelial cells (leading to tissue factor expression). Activation of these other cell types further contributes to the high thrombosis risk accelerated by other factors, such as VWF and procoagulant MP. All of these mechanisms will lead to pituitary apoplexy. (PF4: platelet factor 4; IgG: immunoglobulin G; VITT: vaccine-induced immune thrombotic thrombocytopenia; VWF: Van Willebrand Factor; MP: microparticles).

**Table 1 vaccines-10-02004-t001:** Clinical, hormonal, and radiological presentations of all cases of pituitary apoplexy and hypophysitis following a COVID-19 vaccination.

Reference	Gender	Dose	Vaccine	Symptoms	Days until Symptoms	Diagnostic	Diabetes Insipidus	Thyreotrop Axis	Gonadal Axis	Corticotrop Axis	Somatotrop Axis	Lactotrop Axis	MRI
Roncati et al. (2022) [29]	F	1st	ChAdOx1 nCov-19; AstraZeneca	fever for 24 h and tension-type headache	1 day	Apoplexy	N	N	Hypogonadism	N	N	Hyperprolactinemia	signal alteration related to a hemorrhagic event in the right half of the sella turcica
Zainordin et al. (2022) [28]	F	1st	ChAdOx1 nCov-19; AstraZeneca	severe frontal headache	1 day	Apoplexy	N	N	N	N	N	N	pituitary apoplexy with mass effect to the optic chiasm and cavernous portion
Pinar-Gutiérrez et al. (2021) [30]	F	1st	ChAdOx1 nCov-19; AstraZeneca	Frontal headache	5 days	Apoplexy	N	N	N	N	N	N	adenohypophysis hemorrhagic bleeding in association with a possible 10 mm intraglandular adenoma
Murvelashvili et al. (2021) [35]	M	2nd	BNT162b2; Pfizer-BioNTech	headache, nausea, vomiting, malaise, diffuse arthralgias	3 days	Hypophysitis	+	Hypothyroidism	Hypogonadism	Hypocorticisim	N	N	enlarged pituitary gland consistent with acute hypophysitis
Ankireddypalli et al. (2022) [33]	F	1st	BNT162b2; Pfizer-BioNTech	headache, polydipsia, and polyuria	2 days	Hypophysitis	+	Hypothyroidism	Hypogonadism	N	GH insufficiency	N	thickening of the pituitary stalk
Morita et al. (2022) [31]	M	2nd	BNT162b2; Pfizer-BioNTech	headaches, nausea, and diarrhea	1 day	Hypophysitis	N	N	N	Hypocorticisim	N	N	Atrophic pituitary gland
Bouça et al. (2022) [32]	F	2nd	BNT162b2; Pfizer-BioNTech	intense thirst and polyuria	7 days	Hypophysitis	+	N	N	N	N	N	loss of the posterior pituitary bright spot on T1 weightedimaging
Ach et al. (2022) [34]	F	1st	ChAdOx1 nCov-19; AstraZeneca	headache, polydipsia, and polyuria	3 days	Hypophysitis	+	N	N	N	N	N	thickening of the pituitary stalk

N: normal; F: female; M: male.

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
