# Peer review of "Pilot Findings on SARS-CoV-2 Vaccine-Induced Pituitary Diseases: A Mini Review from Diagnosis to Pathophysiology"

_vaccines, 2022, doi:10.3390/vaccines10122004_

Round 1

Reviewer 1 Report

I reviewed the manuscript entitled " SARS-CoV-2 vaccines-induced pituitary diseases: A mini Review from diagnosis to pathophysiology". I think it is a comprehensively well written review. I have only some minor remarks that the authors should consider.

1.      The abstract is too extensive, I would suggest reducing this part would allow to better highlight the main findings in the review (I suggest to reduce the abstract to a maximum of 400 words).

2.      The sentence “In most 60 countries, both messenger ribonucleic acid (mRNA) and Deoxyribonucleic acid (DNA) 61 vaccines have been administered, and many of their side effects have also been studied” (page 2, line 60) might be too generic. I would suggest to briefly describe, more in detail, the type of available COVID-19 vaccines.

3.      Please uniform table 1 according to the journal style guidelines, since some cell’s content is unclear (i.e., the symptoms of the report of Ankireddypalli and colleagues, or the cell “symptoms” of the report of Ach and colleagues).

4.      Please correct typos and spell check (e.g., in the table 1, “Hyppthyroidism”)

5.      The bibliography can be improved. I would suggest some important references in the field, which can be cited briefly expanding the corresponding section, adding more details in the introduction and discussion sections

-        https://doi.org/10.3390/jcm10132920 and https://doi.org/10.3390/idr14020023, comprehensive reviews on the relationship between COVID-19 and the endocrine system (page 2, line 52)

-        Page 6, line 214: https://doi.org/10.3390/vaccines10071115, a recent large-sample work analysing the adverse effect of the vaccine booster dose; https://doi.org/10.3390/life12091338 and https://doi.org/10.3390/vaccines10040488, recent comprehensive reviews on the possible pathogenesis of severe effects of COVID-19 vaccination

-        https://doi.org/10.3390/ijms23158721, eminent review on the pathogenesis of pituitary apoplexy, (page 9, line 384).

Author Response

Revision Letter

The authors would like to thank the editorial board and the reviewers for all of their careful, constructive and insightful comments in relation to this work. We outline below our responses to these comments, and we indicate where appropriate change has been made in the manuscript as a result. For ease of viewing, the reviewers’ comments are highlighted. We believe that these comments helped greatly in making our manuscript clearer and more useful to the reader.

Editors:

We thank the editors for their diligent reading of our manuscript.

- We checked all references, as suggested by the reviewers.

- We highlighted the revisions as mentioned.

- A cover letter was assigned to the reviewers.

- The article was proofread by two native English speakers.

- The similarities found were corrected and reformulated. Among the many similarities are excerpts from data published previously by some of the authors. We apologize for this redundancy, and assure the editors that all the highlighted repetitions have been corrected.

Reviewer 1:

We thank the reviewer for the excellent grades given to our manuscript. This work has taken a long time to be analyzed and interpreted and we are grateful that it has satisfied his requirements. We thank the reviewer for his constructive comments and suggestions and we discuss these in sequence below.

Changes:

1- The abstract is too extensive, I would suggest reducing this part would allow to better highlight the main findings in the review (I suggest to reduce the abstract to a maximum of 400 words).

Response: The abstract has been reformulated and reduced to less than 400 words as suggested by the reviewer.

  2- The sentence “In most 60 countries, both messenger ribonucleic acid (mRNA) and Deoxyribonucleic acid (DNA) 61 vaccines have been administered, and many of their side effects have also been studied” (page 2, line 60) might be too generic. I would suggest to briefly describe, more in detail, the type of available COVID-19 vaccines.

Response: We agree with the reviewer' suggestion. We have added the names of the vaccines used in the anti-Covid vaccination campaign.

3-  Please uniform table 1 according to the journal style guidelines, since some cell’s content is unclear (i.e., the symptoms of the report of Ankireddypalli and colleagues, or the cell “symptoms” of the report of Ach and colleagues).

Response: We apologize for this technical problem. The table was not, indeed, in the optimal format. A complete rearrangement has been done to improve its visibility.

  1. Please correct typos and spell check (e.g., in the table 1, “Hyppthyroidism”).

Response: Typographical and spelling errors have been corrected.

  1. The bibliography can be improved. I would suggest some important references in the field, which can be cited briefly expanding the corresponding section, adding more details in the introduction and discussion sections

-        https://doi.org/10.3390/jcm10132920 and https://doi.org/10.3390/idr14020023, comprehensive reviews on the relationship between COVID-19 and the endocrine system (page 2, line 52)

-        Page 6, line 214: https://doi.org/10.3390/vaccines10071115, a recent large-sample work analysing the adverse effect of the vaccine booster dose; https://doi.org/10.3390/life12091338 and https://doi.org/10.3390/vaccines10040488, recent comprehensive reviews on the possible pathogenesis of severe effects of COVID-19 vaccination.

-        https://doi.org/10.3390/ijms23158721, eminent review on the pathogenesis of pituitary apoplexy, (page 9, line 384).

Response: All the references suggested by the reviewer have been added.

Reviewer 2:

We thank the reviewer for his comments and suggestions and we discuss these in sequence below.

  1. The primary problem of this article is that the size of the study subject is too low to make any meaningful conclusion. For this reason, the manuscript does not stand on reasonable ground.

Response: We thank the reviewer for his comment. We understand the reviewer's opinion regarding the small number of cases available in literature. However, the purpose of our paper is to shed light on pituitary damage which can go unnoticed, mistaken for a classical post-vaccination syndrome (asthenia, headache) ...

Moreover, analyses on pituitary adverse effects are also rare in other reviews. For instance, in the review addressing the effects of Covid 19 on the pituitary gland (COVID-19 and the pituitary. Pituitary. 2021 Jun;24(3):465-481. doi: 10.1007/s11102-021-01148-1.), we find no more than 10 cases of apoplexy, diabetes insipidus or antepituitary damage. This is explained by the paucity of pituitary lesions which is attributed to the anatomy and the vascularization of the gland and above all, to the overlap of antepituitary insufficiency symptoms with other non-specific symptoms.

In comparison, in reviews on post-vaccination Graves' disease authors collected about 40 cases, because hyperthyroidism is a much more clinically noisy disease(Graves' Disease Following SARS-CoV-2 Vaccination: A Systematic Review. Vaccines (Basel). 2022 Sep 1;10(9):1445. doi: 10.3390/vaccines10091445).

We hope to convince the reviewer of the interest of this paper as an inaugural pillar for other papers to come on post-vaccination pituitary disorders, since these lesions will be better known by doctors on the next vaccination campaigns.

  1. Abstract is too long and tedious to read. The same for the Discussion part. The authors need significant improvement in professional writing arrangement and academic writing experience.

Response: All structures in the article have been reviewed by a native English speaker. The parts suggested by the reviewer have been thoroughly reformulated and restructured.

  1. Why focus on pituitary diseases as a possible consequence of COVID vaccine? The authors do not provide a rationale of this study.

Response:

Covid-19 vaccines have been shown to protect against severe forms of the disease thus reducing mortality among infected patients. All published cases of adverse events were not intended to discuss the value of the vaccine in preventing serious SARS-CoV-2 infections.

However, adverse events, particularly those involving pituitary disorders, should be reported. As a matter of fact, in both hypophysitis and apoplexy (the two major reported forms of pituitary lesions), the prognosis may be compromised due to corticotroph, thyroid hormones or vasopressin deficiencies. Up to date, only ten cases of these side effects have been reported. We strongly believe that the prevalence of these adverse effects is underestimated as they can easily be misdiagnosed as common post-vaccination symptoms such as asthenia, headaches, nausea…These symptoms, in view of a careful clinical examination, can be linked to an inflammation or an ischemia of the pituitary gland (with the intensity of symptoms being correlated to the extent of tissue damage). Furthermore, we have observed that many patients recovered from their post-vaccination symptoms after several weeks, concomitantly to pituitary tissue recovery. The main aim of our paper was to shed light on these symptoms in order to warn clinicians of insidious pituitary’s adverse events.

We are very grateful for all the suggestions made by the reviewers and the editorial board in order to improve our manuscript. We remain available for any further revisions requested.

Reviewer 2 Report

The authors performed a literature review of medical cases of interest and analyzed the adverse effect of SARS-CoV-2 vaccination, mainly in the pituitary system.  The topic is of importance and timely interest, but the presentation of the article is not sound.  Here are some concerns raised:

1. The primary problem of this article is that the size of the study subject is too low to make any meaningful conclusion. For this reason, the manuscript does not stand on reasonable ground.

2. Abstract is too long and tedious to read. The same for the Discussion part. The authors need significant improvement in professional writing arrangement and academic writing experience.

3. Why focus on pituitary diseases as a possible consequence of COVID vaccine? The authors do not provide a rationale of this study.

Author Response

(The authors gave the same response as above.)

Round 2

Reviewer 2 Report

I have reviewed the revision of the authors based on previous comments. For the primary problem as the size of this study is way low, I do not think this is a meaningful conclusion worth publishing. 

Author Response

Minor Revision Letter

The authors would like to thank the editorial board and the reviewers for all of their careful, constructive and insightful comments in relation to this work. We outline below our responses to these comments, and we indicate where appropriate change has been made in the manuscript as a result. For ease of viewing, the reviewers’ comments are highlighted. We believe that these comments helped greatly in making our manuscript clearer and more useful to the reader.

Editors:

We thank the editors for their diligent reading of our manuscript.

- We highlighted the revisions as mentioned.

- A cover letter was assigned to the reviewers.

- We have added to the manuscript three comprehensive figures with a compacted legend, depicting the numerous mechanisms incriminated in the genesis of these pituitary disorders.

The first figure explains the ASIA syndrome and its pathophysiology.

The second figure shows the Physiology of pituitary cell protein's expression and Vaccine induced hypophysitis pathophysiology.

The third figure shows the cascade of pathogenic events that could favor the onset of the vaccine-induced immune thrombotic thrombocytopenia in individuals vaccinated with anti-COVID-19 vaccines.

Reviewer 1:

We thank the reviewer for the excellent grades given to our manuscript. This work has taken a long time to be analyzed and interpreted and we are grateful that it has satisfied his requirements.

Reviewer 2:

We thank the reviewer for his comments and suggestions and we discuss these in sequence below.

  1. The primary problem of this article is that the size of the study subject is too low to make any meaningful conclusion. For this reason, the manuscript does not stand on reasonable ground.

Response: We thank the reviewer for his comments that he developed in the first revision. We have tried on several occasions to explain the perspective of this article. The small sample size might be well explained by the inadvertence of some physicians in the face of nonspecific symptoms that can be attributed to an authentic pituitary insufficiency. We strongly believe that the prevalence of these adverse effects is underestimated as they can easily be misdiagnosed as common post-vaccination symptoms.

The other aim of this manuscript is to clarify the multiple pathophysiological mechanisms of these different disorders already discussed in other reviews. The pituitary lesions remain rare but must be made aware to the physicians in front of this type of vaccination which will be repeated in a seasonal way.

We have further argued our discussion with three figures that will give more clarification on the different pathophysiological mechanisms already mentioned above.

We thank the reviewer for his understanding and patience in reading our manuscript.

We are very grateful for all the suggestions made by the reviewers and the editorial board in order to improve our manuscript.